# Enhanced Osteogenic Differentiation of Human Mesenchymal Stem Cells on Amine-Functionalized Titanium Using Humidified Ammonia Supplied Nonthermal Atmospheric Pressure Plasma

**DOI:** 10.3390/ijms21176085

**Published:** 2020-08-24

**Authors:** Jae-Sung Kwon, Sung-Hwan Choi, Eun Ha Choi, Kwang-Mahn Kim, Paul K. Chu

**Affiliations:** 1Department and Research Institute of Dental Biomaterials and Bioengineering, Yonsei University College of Dentistry, Seoul 03722, Korea; kmkim@yuhs.ac; 2BK21 PLUS Project, Yonsei University College of Dentistry, Seoul 03722, Korea; selfexam@yuhs.ac; 3Department of Orthodontics, Institute of Craniofacial Deformity, Yonsei University College of Dentistry, Seoul 03722, Korea; 4Plasma Bioscience Research Center, Kwangwoon University, Seoul 01897, Korea; ehchoi@kw.ac.kr; 5Department of Physics, Department of Materials Science and Engineering, and Department of Biomedical Engineering, City University of Hong Kong, Kowloon, Hong Kong, China

**Keywords:** atmospheric-pressure plasma, titanium, amine, osteogenic differentiation, mesenchymal stem cells

## Abstract

The surface molecular chemistry, such as amine functionality, of biomaterials plays a crucial role in the osteogenic activity of relevant cells and tissues during hard tissue regeneration. Here, we examined the possibilities of creating amine functionalities on the surface of titanium by using the nonthermal atmospheric pressure plasma jet (NTAPPJ) method with humidified ammonia, and the effects on human mesenchymal stem cell (hMSC) were investigated. Titanium samples were subjected to NTAPPJ treatments using nitrogen (N-P), air (A-P), or humidified ammonia (NA-P) as the plasma gas, while control (C-P) samples were not subjected to plasma treatment. After plasma exposure, all treatment groups showed increased hydrophilicity and had more attached cells than the C-P. Among the plasma-treated samples, the A-P and NA-P showed surface oxygen functionalities and exhibited greater cell proliferation than the C-P and N-P. The NA-P additionally showed surface amine-related functionalities and exhibited a higher level of alkaline phosphatase activity and osteocalcin expression than the other samples. The results can be explained by increases in fibronectin absorption and focal adhesion kinase gene expression on the NA-P samples. These findings suggest that NTAPPJ technology with humidified ammonia as a gas source has clinical potential for hard tissue generation.

## 1. Introduction

Osteoporosis, trauma, and cancer cause damage to and loss of hard tissue, and such conditions are increasing as the baby boomer generation ages [1,2,3,4,5]. Consequently, therapeutic strategies based on hard tissue regeneration that can replace or repair damaged hard tissues are in increasing demand. Over two million hard tissue replacement procedures are currently conducted annually worldwide [3,4,5]. Biomaterials are commonly used in hard tissue regeneration, and titanium (Ti) and Ti alloys are commonly used in orthopedics and dentistry [6,7]. Because of their good biocompatibility, corrosion resistance, and mechanical properties, such as elastic modulus, density, and surface hardness, Ti and Ti alloys have several advantages over other biomaterials in hard tissue regeneration [8]. However, despite recent advances in Ti technology, rapid and firm integration of Ti with surrounding tissue, which is necessary for dental and orthopedic hard tissue regeneration, remains challenging [9]. The integration of biomaterials such as Ti with surrounding bone tissues is commonly known as osseointegration. Osseointegration involves osteoconduction, which enables existing osteoblasts to attach onto the material and proliferate, and osteoinduction, which induces stem cells to differentiate into the osteoblastic lineage [10,11]. Failure or impairment of these two processes results in soft tissue formation surrounding the implant, which causes fibrous encapsulation of the material and prevents achievement of firm fixation over a short period of time [12].

Because of their potential for unlimited proliferation, stem cells are of interest in the development of hard tissue regeneration therapies [2]. Among the various types of stem cells, human mesenchymal stem cells (hMSCs) can be obtained from the bone marrow of patients. These cells are more abundant than embryonic stem cells and present fewer ethical concerns with their extraction. Furthermore, hMSCs represent the precursors of the osteoblastic lineage [13,14] and can also differentiate into other cell lineages, such as chondrocytes [15], myocytes [16], and adipocytes [17]. Differentiation of hMSCs on the surface of biomaterials may be influenced by the properties of the specific biomaterial, such as its elasticity [18] and surface topography [12,19]. Chemical inducing agents, such as dexamethasone and glycerophosphate, have well-known effects on the osteogenic differentiation of hMSCs; however, these agents do not naturally occur in the human body [20]. Recently, the formation of chemical functional groups, such as amines, on biomaterials has been shown to influence hMSC differentiation into the osteoblastic cell lineage [21,22,23]. For example, Curran et al. [22] demonstrated that amino functionalization on a glass coverslip resulted in osteogenic hMSC differentiation, and Wang et al. [23] utilized plasma immersion ion implantation with ammonia gas to modify polytetrafluoroethylene and to induce hMSC differentiation into osteoblasts.

Recently, we reported that the application of NTAPPJ could improve the selected properties of biomaterials without causing thermal damage [24,25]. Compared with low-pressure plasma devices, NTAPPJ technology does not require a vacuum, and it is also cost effective, portable, and easy to use. The effects of NTAPPJ-treated Ti result in the removal of hydrocarbons and enhanced osteogenic activity, similar to the effects achieved by ultraviolet treatment of Ti [26]. Nevertheless, current NTAPPJ technology with supplies of air, oxygen, nitrogen, and argon, etc. is limited in terms of its ability to form specific osteogenic functionalities such as amines.

Herein, we demonstrate the feasibility of guiding hMSC differentiation on Ti via the use of nonthermal atmospheric pressure plasma jet (NTAPPJ) technology, which produces electrons, ions, and free radicals at atmospheric pressure [27]. We developed a specific way to form amine functionalities on the surface of Ti using NTAPPJ supplied with humidified ammonia (Figure 1) following pilot testing with different humidified sources, where samples of titanium were exposed to NTAPPJ with different conditions as outlined in Table 1. We hypothesized that the NTAPPJ-promoted formation of various chemical functionalities on the surface of Ti would affect the differentiation of hMSCs, whereby the amine functionalized surface of Ti by humidified ammonia supplied by NTAPPJ would further enhance the osteogenic activity of hMSCs.

## 2. Results

### 2.1. Morphology and Hydrophilicity of NTAPPJ-Treated Ti Samples

The morphology of the samples was examined by scanning electron microscopy (SEM), and the results showed that a smooth surface with a morphology typical of machined-cut Ti was evident in the control (C-P) group and that the morphology was preserved after plasma exposure for the nitrogen (N-P), air (A-P), and humidified ammonia (NA-P) groups.

Despite the preserved surface morphology, the hydrophilicity of the Ti changed after NTAPPJ exposure. The contact angle on the C-P samples is approximately 75°, indicating a relatively hydrophobic surface, whereas that on the N-P sample is not measurable because of its superhydrophilicity. The A-P and NA-P samples are also relatively hydrophilic compared with the C-P and present contact angles of approximately 25° and 20°, respectively.

Additionally, the changes in contact angle on the Ti surface with respect to time after NTAPPJ exposure have been considered in order to consider reversion of the hydrophilic state of the NTAPPJ-exposed surface (Figure 2). The results showed that N-P, A-P, and NA-P showed reversion in the hydrophilic state with respect to time after initial exposure, though the contact angle values were still lower than C-P at 24 h.

### 2.2. Surface Chemistry of NTAPPJ-Treated Ti Samples

The chemical shifts and subsequent changes in chemical composition induced by the formation of new functional groups on the Ti surface were examined via XPS and are shown in Figure 3. Figure 3a–c displays high-resolution XPS spectra for C1s, O1s, and N1s, respectively, on all samples. Figure 3d shows a histogram of the surface chemistry atomic percentage on each sample calculated from the broad spectrum. Figure 3d shows that O is the most abundant element on the surface of all samples. The C-P treatment showed C as the next most abundant element followed by Ti and very minimal N. In terms of N-P, a decrease in C content was noticeable compared with that of C-P, and the reduction in the C-H/C-C peak (C_2_) is shown in Figure 3a. Although a decrease in the peak corresponding to C-H/C-C (C_2_) was also found for the A-P and NA-P treatments (Figure 3a), the atomic percentage of C was only slightly reduced (Figure 3d) compared with that of the N-P treatment because the A-P and NA-P treatments showed a peak of high binding energy corresponding to the C=O (C_1_) functional group. This shift to a higher binding energy for the two groups was also identified in the O1s spectra, as a shift of the peak from a lower binding energy for the O-Ti (O_2_) functional group to a higher binding energy for the O-H (O_1_) group in both A-P and NA-P is shown in Figure 3b.

In the N1s spectra (Figure 3c), a single peak spectrum corresponding to the N-H (N_2_) functional group was observed for the C-P treatment, and similar spectra were observed for the N-P and A-P treatments. However, a significantly higher intensity of the peak corresponding to the N-H (N_2_) functional group was observed in the NA-P treatment, and peaks exhibiting a high binding energy for N-O (N_1_) and lower binding energy for N-Ti (N_3_) were observed. Indeed, the composition of elemental N was much higher in the NA-P samples compared with any of the other samples (Figure 3d) as the composition has been calculated from the area under curve of XPS results.

### 2.3. Number of Viable Cells and Cell Morphology on the Titanium before and after NTAPPJ Exposure

The viability of the hMSCs in the C-P and test groups was examined under a confocal laser microscope, and the live cells appeared green and the dead cells appeared red (Figure 4a–d). The results indicated that dead red cells were not visible on the surface of either the C-P or test groups, indicating biocompatibility of both the C-P specimen and NTAPPJ-treated Ti. A significantly greater number of green cells were visible for all of the test groups (Figure 4b–d) compared with the C-P (Figure 4a).

The morphology of cells after 4 h of incubation was examined using immunofluorescent images and is shown in Figure 4e–h. A rounded cell morphology with relatively undeveloped actin filaments was evident for the C-P in Figure 4e, whereas stretched cells with actin filaments were evident for N-P (Figure 4f), A-P (Figure 4g), and NA-P (Figure 4h). In terms of quantitative analyses, the results showed that there were no significant differences in number of cell attachments between N-P, A-P, and NA-P (*p* > 0.05) (Figure 5).

### 2.4. Cell Proliferation, Alkaline Phosphatase Acitivity, and Osteogenic Differentiation on Titanium before and after NTAPPJ Exposure

The cell proliferation rate is shown in Figure 6a. Similar to cell attachment, a significantly higher rate of hMSC proliferation was observed for the N-P, A-P, and NA-P groups compared with C-P, although in terms of the differences among the test groups, both A-P and NA-P showed significantly higher levels of proliferation than N-P.

Alkaline Phosphatase (ALP) activity is shown in Figure 6b, and its trend was similar to that for cell proliferation, with significantly higher ALP activity levels observed for the hMSCs in all test groups compared with C-P. In addition, the hMSCs in both the A-P and NA-P groups showed significantly higher ALP activity than the hMSCs in the N-P group. Finally, the NA-P group showed a significantly higher level of ALP activity than the A-P group.

The results of the qRT-PCR analysis for osteogenic markers are shown in Figure 6c–f. The results for ALP gene expression are shown in Figure 6c, and they were similar to the ALP activity results; the N-P, A-P, and NA-P groups all showed significantly higher levels of relative ALP gene expression than C-P, and A-P and NA-P showed significantly higher levels of ALP gene expression than N-P. However, compared with the ALP activity test results, the A-P and NA-P groups did not show differences in terms of the level of relative gene expression. Although significant differences in the relative gene expression of Bone Sialoprotein (BSP) were not observed among the groups (Figure 6d), significantly higher levels of Osteopontin (OPN) expression were observed in N-P, A-P, and NA-P than in C-P and significant differences in expression were not observed among the test groups (Figure 6e). Finally, only NA-P showed a significantly higher level of Osteocalcin (OCN) gene expression compared with C-P (Figure 6f).

### 2.5. Protein Absorption of Titanium and FAK Gene Expression of hMSCs

The amount of Bovine Serum Albumin (BSA) absorption on the surface of the C-P and test groups was measured, and the results for the C-P group were not significantly different compared to those of the test groups (Figure 7a). However, the results for fibronectin absorption were inconsistent with the results for BSA absorption, which was significantly higher in the A-P and NA-P groups than in the C-P and N-P groups (Figure 7b).

The relative expression of Focal Adhesion Kinase (FAK) by hMSCs cultured on the C-P and test samples was measured, and significantly higher levels of expression were observed for the hMSCs cultured on the NA-P sample than those observed in the remaining groups (Figure 7c). Although the level of FAK gene expression was significantly higher for N-P and A-P compared with C-P, significant differences were not observed between the N-P and A-P test groups.

## 3. Discussion

Various molecular functionalities were formed on the surface of the Ti samples, and they influenced on the osteogenic differentiation of hMSCs. Similar results have been reported by other authors who reported the osteogenic guidance of hMSCs on various biomaterials using different chemical functionalities [22,23]. However, this is the first study that considered the use of humid ammonia-supplied NTAPPJ, which forms various chemical functionalities on the surface of Ti that included amine-related functionalities.

Compared with other methods capable of forming osteogenesis-promoting functionalities on biomaterial, the use of NTAPPJ preserved the topographical features of the Ti biomaterial. Topographical changes have been considered a limitation with previous methods of chemical functionalities formation, as such changes would have an additional influence on cellular behavior and tissue formation [28]. This has been shown to be a problem with methods such as magnetron sputtering, physical vapor deposition, or anodization, as the results will cause changes in surface topographical features [7,12,23].

The first change observed after the exposure of Ti to NTAPPJ was an improvement of hydrophilicity. Hydrophilic biomaterials may be favored in dental and medical implants for hard tissue regeneration because the access to blood, which contains necessary cells and proteins, is enhanced [29,30]. The hydrophilicity of Ti changes over time, and aging produces a progressively more hydrophobic surface (Figure 2). This limitation is not solely associated with NTAPPJ technology because other chemical modification techniques exhibit similar time-dependent effects [31,32]. However, this issue is not prohibitive as long as the effects are well understood and addressed during clinical applications; moreover, the portability of the technology offers a unique advantage over other conventional surface treatment techniques because surface treatments via NTAPPJ could occur immediately prior to implant placement. Nonetheless, the samples exposed to NTAPPJ were more hydrophilic than the C-P samples, even after 24 h. Our previous studies also considered samples after NTAPPJ treatment longer than 24 h [33]. The results showed eventual reversion of hydrophilicity to its original state, though the reversions were indicated to be minimized by storage in solution such as distilled water as the method prevented hydrocarbon contamination [33].

Chemical changes to biomaterials often alter the surface hydrophilicity of the material [24]; however, changes in hydrophilicity do not necessarily cause corresponding variations in osteogenic differentiation by hMSCs because osteogenicity is also related to chemical features other than wettability [29]. Accordingly, the chemical changes after plasma exposure were monitored. As described, NTAPPJ produces electrons, radicals, and ions that interact directly or indirectly with the Ti surface. The chemical species produced by NTAPPJ modified the material in terms of chemical composition and formation of new functional groups as indicated by XPS (Figure 3). The formation of chemical functionalities on the surface of Ti can be varied using different gases supplied for NTAPPJ. First, the N-P treatment resulted in a simple reduction of the carbon content compared with C-P. Although this decrease was also observed for the A-P and NA-P treatments, the atomic percentage of carbon was not greatly reduced compared with that of the N-P treatment. The extreme hydrophilicity of N-P and relatively lower hydrophilicity of A-P and NA-P may have been caused by the removal of hydrocarbons and the reduced carbon content on the samples [24,34] because similar results have been achieved by other methods, such as ultraviolet exposure of Ti [29,31,34]. Surface hydrocarbons are formed as a reaction between surface titanium dioxide and the surrounding atmosphere including carbon dioxide [29,31,33]. However, the N-P treatment showed a simple reduction of carbon content without formation of any new functional groups, whereas the A-P and NA-P treatments both formed oxygen-related functional groups, such as C=O or O-H. Finally, the A-P and NA-P samples showed differences in the N1s spectra and N concentration as indicated by a significantly high intensity peak corresponding to the functional group N-H and peaks of high binding energy for the N-O group and lower binding energy for the N-Ti group in the NA-P samples, whereas these peaks were not observed for the A-P samples. It is interesting to note that N-P treatment that used nitrogen as the source of NTAPPJ caused very little changes in surface nitrogen of Ti (Figure 3c). This is perhaps linked to the previously reported study that optical emission spectra analyses of N-P NTAPPJ showed very small amounts of reactive nitrogen species [25,30].

Biological experiments were performed to assess cell viability, attachment, proliferation, and differentiation. The hMSCs showed high levels of cell attachment (Figure 4 and Figure 5) and little to no evidence of cytotoxicity, which are critical characteristics for dental and medical applications [35,36] and consistent with previous studies [24,37,38]. More substantial cell attachment was observed in the N-P, A-P, and NA-P treatments than in C-P (Figure 4 and Figure 5). hMSC attachment is an early event that is essential to the proliferation and differentiation required for bone formation [39]. As previously discussed, topographical differences were not observed between the C-P and test samples; thus, any differences should have been caused by chemical effects introduced by NTAPPJ exposure. Despite the superior hydrophilicity of the N-P samples compared with the A-P and NA-P samples, significant differences (*p* > 0.05) were not observed between the test groups in terms of the number of attached cells, which may have been related to limitations of the in vitro tests. However, this finding provides evidence that cellular attachment is not as strongly correlated with hydrophilic surfaces as previously demonstrated [29].

Similar trends in cell morphology were observed after 4 h (Figure 4), and actin filament formation and stretched cells were observed for all of the test samples. Cell morphology is an important parameter in cell–biomaterial interactions [40]. For example, the high level of actin cytoskeleton formation on the surface of the test groups exposed to NTAPPJ indicates superior cellular perception by the material [41]. However, despite the different chemical changes, obvious differences in cell viability, attachment, or morphology were not observed between the groups exposed to NTAPPJ.

Differences were observed in the proliferation rate and ALP activity (Figure 6a–b) of hMSCs on the plasma-treated samples. Both the A-P and NA-P treatments showed a higher level (*p* < 0.05) of proliferation than the N-P treatment, and the NA-P showed higher levels (*p* < 0.05) of ALP activity than A-P. Proliferation of hMSCs may indicate osteoconductive properties of the material because it is consistent with the number of mature or progenitor osteoblast cells derived from hMSCs [42]. These features may be modified by changes in the surface chemistry induced by NTAPPJ or similar treatments [30,37,38,43]. The observed chemical changes improved the cell proliferation of the test groups relative to C-P, and the presence of chemical functionality, such as C=O and O-H groups, further enhanced the effects as shown for the A-P and NA-P treatments. This finding highlights the advantage of NTAPPJ over other technologies, such as ultraviolet treatments for dental implants, because these treatments reduce only the carbon content but do not form other functional groups [29,31,34]. A reduction in the carbon content enhances cell attachment and morphological transformation but has limited effects on the proliferation rates in the absence of the necessary bioactive chemical functional groups.

ALP is a key enzyme involved in osteogenic cell formation, and it regulates phosphatase metabolism via the hydrolysis of phosphate esters [44]. The improved ALP activity therefore indicates increased osteogenic activity. Here, greater osteogenic activity was observed from all test groups, with the largest enhancement shown by the NA-P group, which possesses more nitrogen-related functionalities, such as N-H, N-O, and N-Ti; is linked to the osteogenic differentiation of hMSCs on the Ti samples; and is related to successful osteoinduction and early osseointegration. Four primers were considered in the quantitative real-time polymerase chain reaction (qPCR) amplification of ALP, BSP, OPN, and OCN (Figure 6c–f). High BSP expression indicates the formation of proteins that bind strongly to hydroxyapatite via a negatively charged domain, thereby resulting in bone formation [45]. OPN expression is related to the formation of a phosphoprotein that serves as the bridge between osteoblasts and hydroxyapatite [46]. OCN is a calcium-binding protein that regulates bone crystal growth and serves as the most specific marker for osteoblast maturation [47,48]. To investigate the effects of NTAPPJ exposure on the chemical functional groups while minimizing other influences, the cells were cultured in basic medium without osteogenic supplements such as dexamethasone or glycerophosphate. A higher level of ALP and OPN gene expression was observed in each test group than in C-P (*p* < 0.05). However, only NA-P showed a significantly higher level (*p* < 0.05) of OCN gene expression than C-P. Because OCN represents the most specific marker in osteoblast maturation, these findings indicate that the NA-P samples were the most potent in terms of osteogenic differentiation of hMSCs. The difference between the NA-P and the C-P and other test groups results from the presence of nitrogen-related functional groups, such as N-O, N-H, and N-Ti, which favor osteogenic differentiation of stem cells [49] because the positive charge of the chemical functional groups increases affinity for fibronectin [50,51].

To study the possible mechanisms underlying osteogenic differentiation and variations in hMSC differentiation, the absorption of fibronectin and BSA was examined (Figure 7a,b). The results show that greater fibronectin absorption (*p* < 0.05) occurred on the NA-P samples than on the C-P and N-P samples, whereas differences in BSA absorption (*p* > 0.05) were not observed between the C-P and test groups. BSA carries a neutral charge and is less influenced by the charge on the surface of the biomaterial [52]. However, because fibronectin carries negative charges, absorption of this protein is influenced by the positive charge on the biomaterial generated by the various functional groups [52].

The presence of fibronectin and the subsequent exposure of receptors that bind to specific integrin receptors increase the expression of osteogenic differentiation genes via the FAK-ERK pathway [53,54]. Therefore, the absorption of fibronectin is an important initial step in the osteogenic differentiation of hMSCs, and it was investigated by qRT-PCR analyses of FAK gene expression in the hMSCs cultured on the samples for 6 h in a basic medium (Figure 7c). The results show a significantly higher level (*p* < 0.05) of FAK gene expression in the hMSCs cultured on the NA-P samples than in those cultured on the other samples, which corroborates the hypothesis. However, the data do not explain all observed phenomena. As observed from the NA-P group, a substantially higher level (*p* < 0.05) of fibronectin absorption occurred on the A-P samples than on the C-P or N-P samples, and significant differences (*p* > 0.05) were not observed between the NA-P and A-P samples. Nonetheless, the hMSCs cultured on the A-P samples showed a significantly lower level (*p* < 0.05) of FAK gene expression than those cultured on the NA-P samples.

The possible mechanism underlying these differences is summarized in Figure 8. The C-P samples that were not subjected to NTAPPJ treatment possessing hydrocarbons exhibited a lower degree of cell attachment and a more rounded cellular morphology relative to the hMSCs on the plasma-exposed samples. NTAPPJ removes hydrocarbons and increases cell attachment, and the attached cells appear stretched. Oxygen-related chemical functional groups, such as OH, were formed on the A-P and NA-P samples, which resulted in more absorption of fibronectin and consequently higher levels of hMSC proliferation and ALP activity on the A-P and NA-P samples than on the N-P samples. Differences between the A-P and NA-P groups were observed because nitrogen-related functional groups may play an additional role in enhancing FAK gene expression and osteogenic differentiation.

Despite the limitations associated with in vitro studies and the need for further in vivo or other related research, the present findings may provide new insights for developing Ti implants with a higher level of osseointegration using humid ammonia-supplied NTAPPJ. Future studies are currently planned to determine possible clinical applications of the technology as a bioactive surface treatment for biomaterials, including Ti.

## 4. Materials and Methods

### 4.1. Sample Preparation

Machine-cut, grade IV commercially pure titanium (cp-Ti) was used. Round specimens with a diameter of 12 mm and a thickness of 1 mm were cleaned ultrasonically with acetone (99.5%, Duksan Pure Chemical, Ansan, Korea), ethanol (99.9%, Duksan Pure Chemical, Ansan, Korea), and distilled water sequentially for 5 min each at room temperature. The samples were dried naturally in air at room temperature before exposure to NTAPPJ. The specimens for the cellular experiments were sterilized in an autoclave at 121 °C for 15 min.

### 4.2. NTAPPJ Treatment

NTAPPJ was developed by the Plasma Bioscience Research Center (PBRC, Kwangwoon University, Seoul, Korea). The maximum discharge voltage and discharge current were 2.24 kV and 1.08 mA, respectively. Discharge occurred between the inner and outer electrodes of the NTAPPJ system, with porous alumina being the dielectric between the two electrodes (Figure 1a). The gases were supplied to the NTAPPJ system through mass flow controllers (MFC; AFC500, ATOVAC, Yongin-si, Korea) at a flow rate of 1 L/min (Figure 1b). Three different gases were used: nitrogen, air (compressed air), and humidified ammonia. In the case of humidified ammonia, special settings were considered involving the passage of nitrogen through the 0.1 M ammonia solution (Duksan Pure Chemical, Ansan, Korea) at a rate of 1 L/min (Figure 1b). The gas flow was controlled such that the ratio of nitrogen to humidified ammonia was 1 to 9. The control group was not exposed to NTAPPJ. The titanium specimens were placed 10 mm away from the NTAPPJ system (Figure 1a,b), and the plasma exposure time was 4 min. The NTAPPJ operation conditions are summarized in Table 1.

### 4.3. Surface Characterization

The surface morphology of the control and test samples was examined by SEM (JSM-6700F, JEOL, Japan) at 20 kV and 5000× magnification. The specimens were coated with platinum prior to SEM examination. The wettability of the test and control groups was measured by monitoring the water contact angles in accordance with previous studies [24,30,33]. A volume of 8 µL of distilled water was used for measurements, and the contact angle was measured after 30 s at room temperature on a video contact angle measurement system (Phoenix 300, SEO, Suwon, Korea). The change in wettability at 1, 2, 4, 12, and 24 h after NTAPPJ exposure was monitored. The surface chemical composition was determined by X-ray photoelectron spectroscopy (XPS; K-alpha, Thermo VG Scientific, Waltham, MA, USA). Monochromatic Al Kα was used as the X-ray source (Al Kα line: 1486.6 eV), and the sampling area was 400 µm. The spectra were recorded with a pass energy of 200 eV (step size of 1.0 eV) in survey mode and 50 eV (step size of 0.1 eV) in the high-resolution mode to acquire the C1s, O1s, N1s, and Ti2p spectra with a resolution of 0.78 eV measured from the Ag 3d5/2 peaks. The binding energies were referenced to the C1s peak at 284.8 eV.

### 4.4. hMSC Cell Culture

The hMSCs were purchased commercially (Lonza, Allendale, NJ, USA). Cells were passaged three to five times to maintain multilineage capabilities. The cells were cultured on a 100-mm-diameter culture dish (SPL, Daegu, Korea) at 37 °C and 5% CO_2_ in an incubator. Dulbecco’s modified Eagle’s medium (DMEM, Gibco, Grand Island, NY, USA) supplemented with 10% human fetal bovine serum (FBS; Gibco, Grand Island, NY, USA) and 1% antibiotic-antimycotic (Gibco, Grand Island, NY, USA) were used without the addition of any osteogenic differentiation-inducing supplements.

### 4.5. Cell Viability Assay

The viability of hMSCs cultured on the test and control samples was examined using a calcein and ethidium homodimer-1-based staining kit (LIVE/DEADTM Viability/Cytotoxicity Kit, Invitrogen Co., Eugene, OR, USA). A total of 5 × 10^4^ cells was placed on the samples, transferred to 12-well culture plates (SPL, Daegu, Korea), and cultured in an incubator at 37 °C and 5% CO_2_ for 24 h. The cell culture medium was removed from each well, and the cells were washed with phosphate-buffered saline (PBS; Gibco, Grand Island, NY, USA). The stock solution mixed with 10 mL of PBS was applied to the cells for 45 min at room temperature. The cells on each specimen were examined under a confocal laser microscope (LSM700, Carl-Zeiss, Thornwood, NY, USA) with live cells appearing green and dead cells appearing red.

### 4.6. Cell Morphology

A total of 5 × 10^4^ cells was placed on the samples, transferred to a 12-well culture plate (SPL, Daegu, Korea), and cultured in an incubator at 37 °C and 5% CO_2_ for 4 h. The unattached cells were washed away with phosphate buffered saline (PBS; Gibco, Grand Island, NY, USA). The morphology of the attached cells was assessed using fluorescent dyes and a confocal laser microscope. After culturing for 4 h under the previously described conditions, the cells were stained with DAPI (4′,6-diamidino-2-phenylindole, blue coloration for nuclei, Invitrogen, Grand Island, NY, USA) and rhodamine phalloidin (red coloration for actin filament, Invitrogen, Grand Island, NY, USA) and visualized under a confocal laser microscope (LSM700, Carl-Zeiss, Thornwood, NY, USA).

### 4.7. Cell Proliferation Assay

The proliferation of hMSCs was evaluated using BrdU incorporation during DNA synthesis. A total of 5 × 10^4^ cells was placed on each sample, transferred to a 12-well culture plate (SPL, Daegu, Korea), and cultured in an incubator at 37 °C and 5% CO_2_. After 3 days, 100 µL of 100 mM BrdU solution (Roche Applied Science, Penzberg, Germany) was added to each culture well, and the cells with incorporated BrdU were further incubated for 12 h. The cells with incorporated BrdU reacted with anti-BrdU conjugated with peroxidase (Roche Applied Science, Penzberg, Germany) for 90 min. Finally, tetramethylbenzidine (Roche Applied Science, Penzberg, Germany) was added to enable color development. The optical absorbance was measured on a reader (Epoch, BioTek Instruments, Winooski, VT, USA) at 370 nm, and the rate of proliferation was expressed as the percentage of that of the control, C-P.

### 4.8. Alkaline Phosphatase Activity Assay

The ALP activity of hMSCs cultured on each sample for 14 days was examined using a Sensolyte^®®^ p-nitrophenyl phosphate (pNPP) ALP assay kit (AnaSpec, San Jose, CA, USA). Briefly, 1 × 10^4^ cells were placed on the surface of each sample, placed in a 12-well culture plate (SPL, Daegu, Korea), and cultured in an incubator at 37 °C and 5% CO_2_. After 14 days, the cells on each sample were washed twice with PBS and lysed with 1 mL of 0.2% Triton X-100 (AnaSpec, San Jose, CA, USA). The lysed products were further incubated at 4 °C for 10 min under agitation, and the cell suspension was centrifuged at 2500× *g* for 10 min at 4 °C. Fifty microliters of the supernatant weree reacted with 50 μL of the pNPP assay buffer, and the absorbance was determined at 405 nm on an absorbance reader (Epoch, BioTek Instruments, Winooski, VT, USA) after 2 h. The quantity of ALP was determined by allowing 50 μL of ALP with the standard concentration to react with 50 μL of the pNPP assay buffer, and a calibration curve was derived. Total protein was quantified using a microbicinchoninic acid (MicroBCA)-based total protein assay kit (Thermo Fisher Scientific, Rockford, IL, USA) according to the manufacturer’s instructions. Briefly, the supernatant (150 μL) obtained from the previous centrifugation step was mixed with 150 μL of the working solution from the kit and incubated at 37 °C for 2 h. The optical absorbance was measured at 562 nm on a plate reader (Epoch, BioTek Instruments, Winooski, VT, USA). As described previously, the quantity of total protein was determined by reacting 150 μL of bovine serum albumin with 150 μL of the working solution and a calibration curve was generated. The ALP activity was expressed as the amount of quantified ALP per the amount of quantified total protein.

### 4.9. Gene Expression Analysis

The osteogenic gene expression of hMSCs on each sample was determined by qRT-PCR. A total of 1 × 10^4^ cells was placed on each sample, transferred to each well of a 12-well culture plate (SPL, Daegu, Korea), and cultured in an incubator at 37 °C and 5% CO_2_ for 14 days. To analyze FAK gene expression, 5 × 10^4^ cells were placed on each sample in a 12-well culture plate (SPL, Daegu, Korea). The cells were cultured in an incubator at 37 °C and 5% CO_2_ for 12 h. After the incubation periods (14 days for osteogenic gene expression and 12 h for FAK gene expression), the culture media were removed and the cells were washed with PBS (Gibco, Grand Island, NY, USA). Total RNA was extracted from the cells using TrizolTM reagent (Sigma-Aldrich, St. Louis, MO, USA), and cDNA was prepared using a high capacity RNA-to-cDNA kit (Applied Biosystems, Foster City, CA, USA). qRT-PCR was performed with cDNA using SYBR green dye (Applied Biosystems, Foster City, CA, USA), with primers corresponding to ALP, BSP, OCN, and OPN for osteogenic gene expression and to FAK for FAK gene expression [30]. Glyceraldehyde-3-phosphate dehydrogenase (GAPDH) was used as a housekeeping reference gene in both experiments. Quantification was carried out on the PCR system (7300 Real-Time PCR System, Applied Biosystems, Foster City, CA, USA) using the following cycle protocol: 1 cycle at 50 °C for 2 min, 1 cycle at 95 °C for 10 min, 40 cycles at 95 °C for 15 s, and 1 cycle at 60 °C for 1 min. The expression of each osteogenic gene was expressed as the CT (cycle threshold) value relative to that of the control (C-P). All experiments were repeated three times.

### 4.10. Protein Absorption Assay

The absorption of two proteins, BSA and fibronectin, on each sample was investigated. BSA (Pierce Biotechnology, Inc., Rockford, IL, USA) and fibronectin (Sigma-Aldrich, St. Louis, MO, USA) solutions were prepared by mixing the corresponding protein powders with PBS (1 mg of protein per 1 mL of PBS). In each group, 300 μL of the protein solution was pipetted and spread over a titanium disk. After incubation for 24 h at 37 °C, the non-adherent proteins were removed and the amounts of each protein were measured by adding 300 μL of the microbicinchoninic acid (MicroBCA)-based total protein assay kit (Thermo Fisher Scientific, Rockford, IL, USA). The samples were further incubated at 37 °C for 2 h, and the optical absorbance was measured at 562 nm on a plate reader (Epoch, BioTek Instruments, Winooski, VT, USA).

### 4.11. Statistical Analysis

All statistical analyses were performed using IBM SPSS soft-ware, version 23.0 (IBM Korea Inc., Seoul, Korea) for Windows. The biological data are expressed as the mean ± S.D. of at least three independent experiments. Statistical significance was evaluated by one-way analysis of variance with Tukey’s post hoc test. Values of *p* < 0.05 were considered statistically significant.

## 5. Conclusions

The present work demonstrates the advantages of using NTAPPJ technology with respect to the generation of bioactive functional groups on Ti. In particular, the use of humid ammonia-supplied NTAPPJ resulted in the formation of amine chemical functional groups on Ti without changing the topography of the material. Despite the limitations associated with in vitro studies and the need for further in vivo or other related research, the present findings provide new insights for developing Ti implants with a higher level of osseointegration. Future studies are currently planned to determine possible clinical applications of the technology as a bioactive surface treatment for biomaterials, including Ti. Nevertheless, it was evident that different plasma gases had different effects on osteogenic differentiation. NA-P treatment generated nitrogen-related functional groups and led to a higher level of osteogenic differentiation in the corresponding hMSCs. Thus, we discovered a novel surface treatment method for Ti using NTAPPJ with NA-P that can enhance the osteoconductive and osteoinductive activities of Ti.

## Figures and Tables

**Figure 1 ijms-21-06085-f001:**
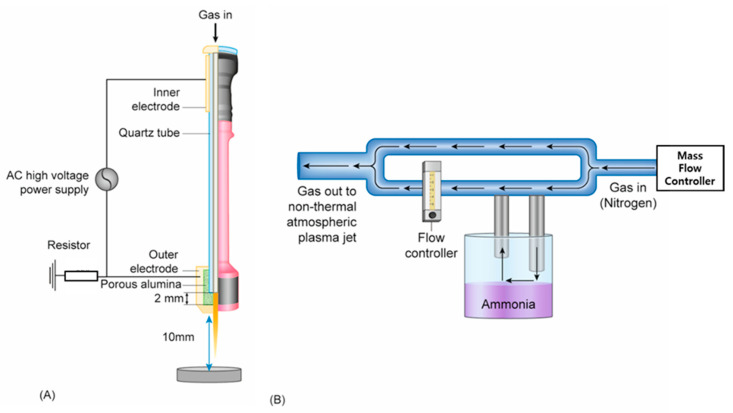
(**A**) Schematic diagram of the nonthermal atmospheric pressure plasma jet and (**B**). the design for a customized device that would supply nitrogen mixed with humidified ammonia.

**Figure 2 ijms-21-06085-f002:**
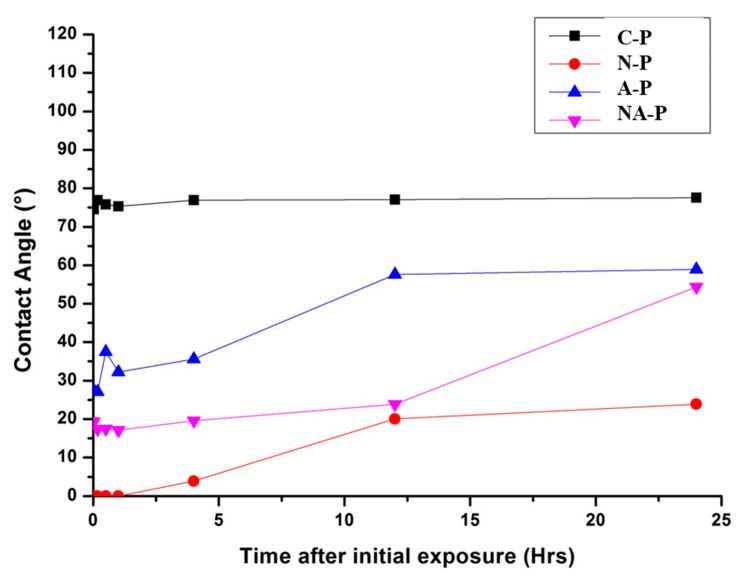
Water contact angles on the samples with time after exposure to nonthermal atmospheric pressure plasma jet.

**Figure 3 ijms-21-06085-f003:**
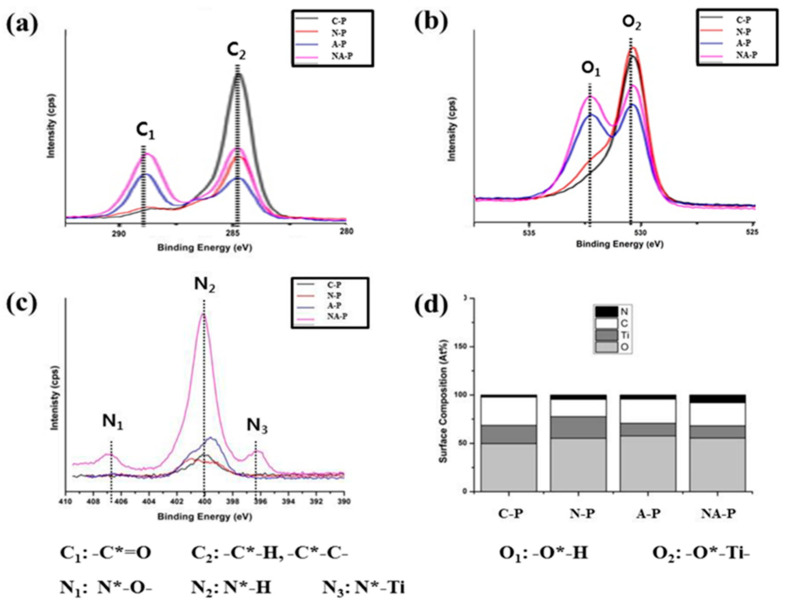
Surface chemistry of the samples as indicated by high-resolution spectra: (**a**) C1s, (**b**) O1s, and (**c**) N1s. The text below the figure refers to the contribution of each peak. (**d**) Atomic percentages. Each relevant element analyzed is marked with *.

**Figure 4 ijms-21-06085-f004:**
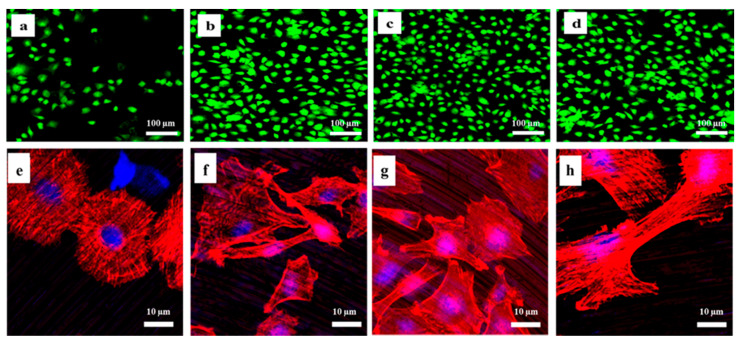
Immunofluorescence images of attached live mesenchymal stem cells: (**a**) control (C-P), (**b**) nitrogen (N-P), (**c**) air (A-P), and (**d**) humidified ammonia (NA-P) (scale bar: 100 μm). Immunofluorescence images of the cytoskeleton of the mesenchymal stem cells (MSCs): (**e**) C-P, (**f**) N-P, (**g**) A-P, and (**h**) NA-P (scale bar: 10 μm). The red color indicates actin filaments, and the blue color shows the nuclei.

**Figure 5 ijms-21-06085-f005:**
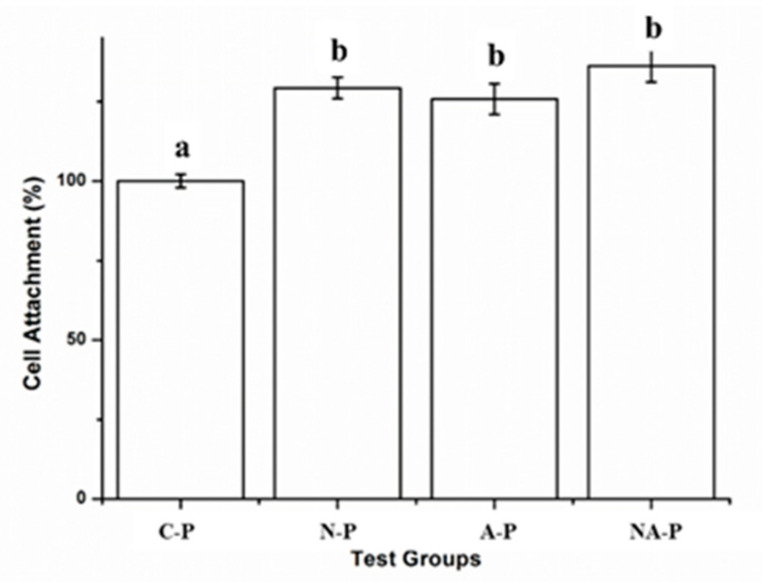
Cell attachment on control and test groups after 4 h: The same lowercase letter indicates no significant differences (*p* > 0.05), and different lowercase letters indicate significant differences (*p* < 0.05), where the greater cellular attachment is higher in alphabetical order (i.e., “b” is significantly greater than “a” and so on).

**Figure 6 ijms-21-06085-f006:**
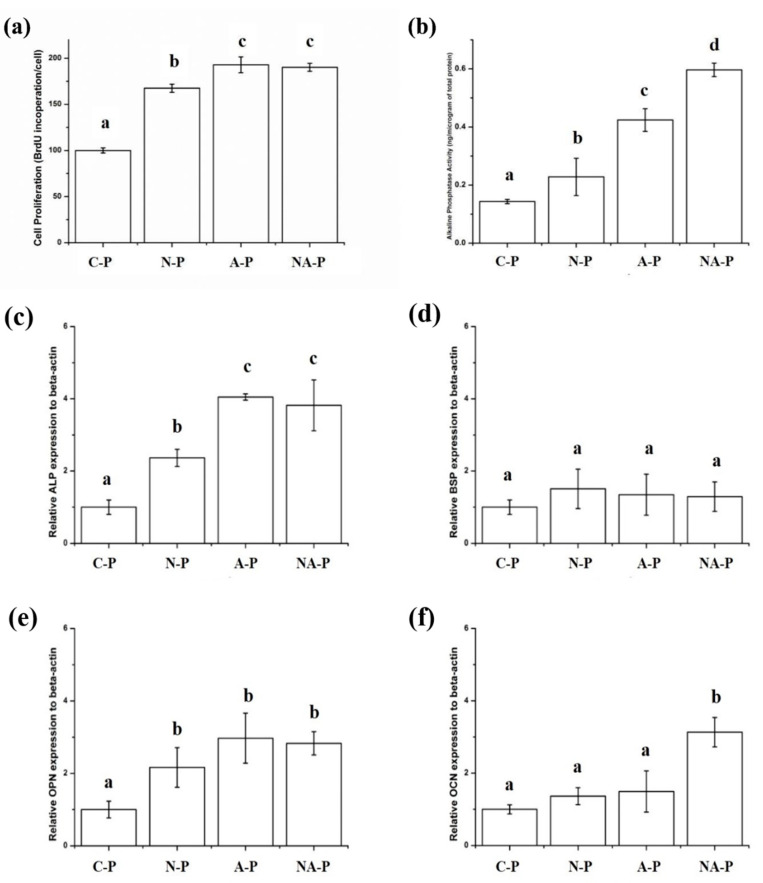
(**a**) Cell proliferation rate and (**b**) alkaline phosphatase activity of human mesenchymal stem cells (hMSCs) cultured on the samples and osteogenic differentiation of hMSCs on the surface of the control and test groups analyzed by qPCR for osteogenic markers: (**c**) alkaline phosphatase (ALP), (**d**) bone sialoprotein (BSP), (**e**) osteopontin (OPN), and (**f**) osteocalcin (OCN) after culturing for 14 days. All the results are expressed relative to the gene expression of C-P as the control. The same lowercase letter indicates no significant difference (*p* > 0.05), and different lowercase letters indicate significant differences (*p* < 0.05). The greater the cellular proliferation or alkaline phosphatase activity or relative gene expression, the higher the letter in alphabetical order (i.e., “b” is significantly greater than “a”, and so on).

**Figure 7 ijms-21-06085-f007:**
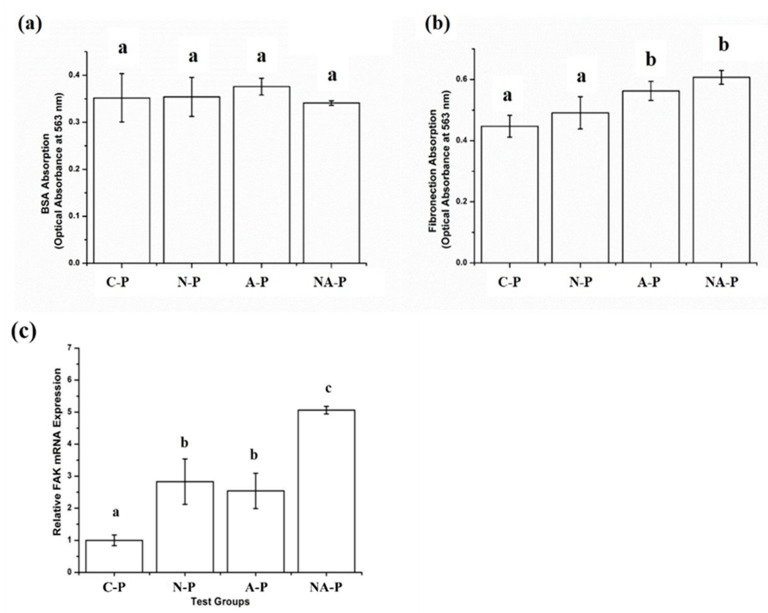
Protein absorption in the control and test groups for two different proteins: (**a**) bovine serum albumin and (**b**) fibronectin. (**c**) Relative gene expression level of focal adhesion kinase (FAK) in hMSCs cultured on the control and test groups where all the results are expressed as relative to gene expression of C-P as the control: The same lowercase letter indicates no significant difference (*p* > 0.05), and different lowercase letters indicate significant differences (*p* < 0.05). The higher the protein absorption or higher the relative gene expression, the higher the letter in alphabetical order (i.e., “b” is significantly greater than “a” and so on).

**Figure 8 ijms-21-06085-f008:**
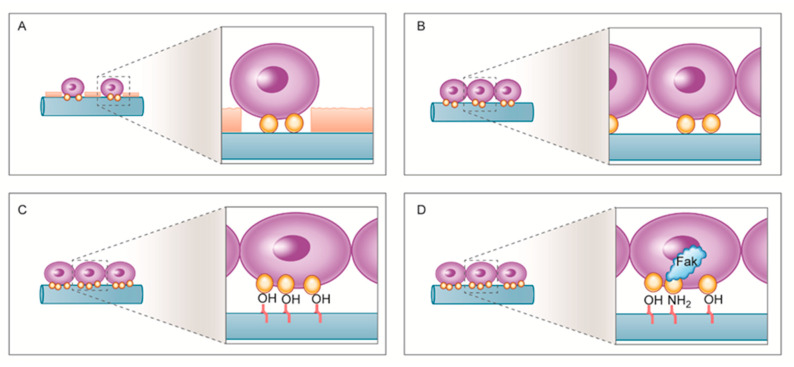
Schematic drawing of the putative mechanism underlying interactions between hMSCs and the (**A**) C-P, (**B**) N-P, (**C**) A-P, and (**D**) NA-P groups: For the N-P, A-P, and NA-P groups, more cell attachment was observed due to removal of hydrocarbon molecules from C-P. Higher levels of fibronectin absorption were observed on the A-P and NA-P groups, likely due to the presence of oxygen-related functional groups, which enhance osteogenic differentiation. The presence of nitrogen-related functional groups is considered to increase focal adhesion kinase (FAK) gene expression, further enhancing osteogenic differentiation on the NA-P samples.

**Table 1 ijms-21-06085-t001:** Nonthermal atmospheric pressure plasma jet exposure conditions and assignment of sample codes.

	Gas Sources	Gas Flow Rate (L/min)	Voltage (kV)	Current (mA)	Treatment Duration (min)	Sample Code
Control	No gas sources	N/A	N/A	N/A	0	C-P
Test Groups	Air (compressed)	1	2.24	1.08	4	A-P
	Nitrogen	1	2.24	1.08	4	N-P
	Nitrogen/Ammonia (humidified)	1	2.24	1.08	4	NA-P

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
