# Peer review of "Enhanced Osteogenic Differentiation of Human Mesenchymal Stem Cells on Amine-Functionalized Titanium Using Humidified Ammonia Supplied Nonthermal Atmospheric Pressure Plasma"

_ijms, 2020, doi:10.3390/ijms21176085_

Round 1

Reviewer 1 Report

Review of Kwon et al, ‘Enhanced osteogenic differentiation of human mesenchymal stem cells on amine-functionalized titanium using humidified ammonia supplied non thermal atmospheric pressure plasma’. This manuscript is a very interesting study on the modification of Ti surface characteristics through cold plasma using different types of gases. The authors demonstrate how changes in surface chemistry result in better cell adhesion and differences in osteogenic differentiation and provide a plausible hypothesis for the underlying mechanism. The implications for use of this technology in implant technology are exciting and the study provides valuable insight into mechanisms of modulating hMSC differentiation. The manuscript is very well-written and clearly presented. I have only few queries outlined below and recommend minor revisions prior to publication.

  • In Line 233 the authors refer to the removal of hydrocarbons from the Ti surface whereas in other sections of the manuscript they refer to carbohydrates. As these are different classes of compounds, the authors should be consistent and precise. Also some explanations of the causes of these impurities on Ti surfaces should be provided e.g. in the introduction.

  • The authors tested air, nitrogen and nitrogen/humidified ammonia. Due to this experimental set-up it is, however, not completely distinguishable if the superior behaviour of NA-P samples is a result of the ammonia or the moisture or both. Why did the authors not use the same experimental set-up to test humidified air and humidified nitrogen which could provide more clarity?

  • Figure 3c shows very little changes in surface nitrogen for the N-AP. Can the authors comment on this, please.

  • What does superscript a in table 1 stand for? (Nitrogen/Ammonia (humidified)a))

  • Line 207 Compared with other methods capable of forming osteogenesis-promoting functionalities on biomaterial, the use of NTAPPJ preserved the topographical features of the Ti biomaterial (Fig. 2a-d). Topographical changes have been considered a limitation with previous methods of chemical functionalities formation, as such changes would have an additional influence on cellular behaviour and tissue formation [28].

Some more details/examples on the ‘other methods’ the authors refer to would be helpful in the discussion.

  • line 215 The hydrophilicity of Ti changes over time, and ageing produces a progressively more hydrophobic surface (Fig. 2f). line 221 Nonetheless, the samples exposed to NTAPPJ were more hydrophilic than the C-P samples, even after 24 hours.

Have the authors investigated the hydrophilicity at time points greater than 24 hours? Does the hydrophilicity revert to that of the untreated sample over time? What are the possible explanations for the loss of hydrophilicity?

  • Line 233 The extreme hydrophilicity of the N-P and relatively low hydrophilicity of the A-P and NA-P may have been caused by the removal of hydrocarbons and the reduced carbon content on the samples [24, 33] …

I’m not sure this sentence is phrased correctly: the hydrophilicity of A-P and NA-P still appears high even if it was lower than for N-P. Do the authors mean to say … the relatively lower hydrophilicity of the A-P and NA-P?

  • Despite the superior hydrophilicity of the N-P samples compared with the A-P and NA-P samples, significant differences (p > 0.05) were not observed between the test groups in terms of the number of attached cells, which may have been related to limitations of the in vitro tests.

Which quantitative data does this refer to? Fig. 4 only shows qualitative data (images) of cell attachment. Please clarify how these images were quantitatively evaluated and show the relevant data.

  • The sequence of primers used in the qRT-PCR should be provided or referenced accordingly.

Minor spelling mistakes

- 2.4. Cell Proliferation, Alkaline Phosphatase Acitivy, and Osteogenic Differentiation on on Titanium Before and After NTAPPJ Exposure

= 2.4. Cell Proliferation, Alkaline Phosphatase Activity, and Osteogenic Differentiation on Titanium Before and After NTAPPJ Exposure

  • The C-P samples that were not subjected to the NTAPPJ treatment possess carbohydrates exhibited a lower degree of..

= The C-P samples that were not subjected to the NTAPPJ treatment and possess carbohydrates exhibited a lower degree of…

  • Therefore, the absorption of fibronectin is an important initial step in the osteogenic differentiation of hMSCs, and it was investigated by via qRT-PCR analyses of FAK gene expression in the hMSCs cultured on the samples for 6 hours in basic medium

= either: … investigated by qRT-PCR… or …investigated via qRT-PCR…

Author Response

Reviewer #1

General comment

Review of Kwon et al, ‘Enhanced osteogenic differentiation of human mesenchymal stem cells on amine-functionalized titanium using humidified ammonia supplied non thermal atmospheric pressure plasma’. This manuscript is a very interesting study on the modification of Ti surface characteristics through cold plasma using different types of gases. The authors demonstrate how changes in surface chemistry result in better cell adhesion and differences in osteogenic differentiation and provide a plausible hypothesis for the underlying mechanism. The implications for use of this technology in implant technology are exciting and the study provides valuable insight into mechanisms of modulating hMSC differentiation. The manuscript is very well-written and clearly presented. I have only few queries outlined below and recommend minor revisions prior to publication.

Response to General Comment

Thank you for your kind and encouraging comments. We now have carried out revisions in accordance to your recommendation and hope that the manuscript is now suitable for publications.

Comment 1

In Line 233 the authors refer to the removal of hydrocarbons from the Ti surface whereas in other sections of the manuscript they refer to carbohydrates. As these are different classes of compounds, the authors should be consistent and precise. Also some explanations of the causes of these impurities on Ti surfaces should be provided e.g. in the introduction.

Response to Comment 1

We now have modified wording so that the word hydrocarbon is consistent. Also we have included following sentences to indicate the causes of these impurities on Ti surfaces in Discussion.

Surface hydrocarbons are formed as reaction between surface titanium dioxide and surrounding atmosphere including carbon dioxide [29, 31, 33].

Comment 2

The authors tested air, nitrogen and nitrogen/humidified ammonia. Due to this experimental set-up it is, however, not completely distinguishable if the superior behaviour of NA-P samples is a result of the ammonia or the moisture or both. Why did the authors not use the same experimental set-up to test humidified air and humidified nitrogen which could provide more clarity?

Response to Comment 2

The experimental set-up was based on our preliminary pilot study where the results were not included in the manuscript. We have considered other humidified sources, that are available in liquid form. Liquid nitrogen was difficult to be humified, but others were possible where the results did not produce any significant improvements. Following sentence is now added;

We developed a specific way to form amine functionalities on the surface of Ti using NTAPPJ supplied with humidified ammonia (Fig. 1), following pilot testing with different humidified sources, where samples of titanium were exposed to NTAPPJ with different conditions outlined in Table 1

Comment 3

Figure 3c shows very little changes in surface nitrogen for the N-AP. Can the authors comment on this, please.

Response to Comment 3

Thank you for your question and we think the comment is related to N-P and not NA-P as NA-P resulted in significant changes in nitrogen as seen in Fig 3c. We have several papers that used nitrogen as the source of plasma gas, and tested on titanium surfaces. All of the results were consistent as it did not results in significant changes in surface nitrogen. In our previous papers, it has been indicated that the results are perhaps linked to species produced by plasma. Analyses such as optical emission spectra showed that nitrogen source did not produce much of reactive nitrogen species, and rather acted as other noble gases (such as argon etc). Following sentence is now added in Discussion;

It is interesting to note that N-P treatment that used nitrogen as the source of NTAPPJ caused very little changes in surface nitrogen of Ti (Fig. 3c). This is perhaps linked to the previously reported study that optical emission spectra analyses of N-P NTAPPJ showed very small amount of reactive nitrogen species [25, 30, 36].

Comment 4

Line 207 Compared with other methods capable of forming osteogenesis-promoting functionalities on biomaterial, the use of NTAPPJ preserved the topographical features of the Ti biomaterial (Fig. 2a-d). Topographical changes have been considered a limitation with previous methods of chemical functionalities formation, as such changes would have an additional influence on cellular behaviour and tissue formation [28].

Some more details/examples on the ‘other methods’ the authors refer to would be helpful in the discussion.

Response to Comment 4

Thank you for your valuable input and suggestions. We not have listed few examples of other methods as below;

This has been shown to be a problem with methods such as magnetron sputtering, physical vapor deposition or anodization, as the results will cause changes in surface topographical features [30, 39, 53].

Comment 5

line 215 The hydrophilicity of Ti changes over time, and ageing produces a progressively more hydrophobic surface (Fig. 2f). line 221 Nonetheless, the samples exposed to NTAPPJ were more hydrophilic than the C-P samples, even after 24 hours.

Have the authors investigated the hydrophilicity at time points greater than 24 hours? Does the hydrophilicity revert to that of the untreated sample over time? What are the possible explanations for the loss of hydrophilicity?

Response to Comment 5

We have considered changes in hydrophilicity over long period time in our different study. The revert to original state is thought to be linked with exposure to surrounding atmosphere with reaction with surrounding gas, as the previous study indicated that storage method would result in significant changes in hydrophilicity revert. Following is now added;

Our previous studies also considered samples after NTAPPJ treatment longer than 24 hours [33]. The results showed eventual revert of hydrophilicity to original state, though the reverts were indicated to be minimized by storage in solution such as distilled water as the method prevented hydrocarbon contamination [33].

Comment 6

Line 233 The extreme hydrophilicity of the N-P and relatively low hydrophilicity of the A-P and NA-P may have been caused by the removal of hydrocarbons and the reduced carbon content on the samples [24, 33] …

I’m not sure this sentence is phrased correctly: the hydrophilicity of A-P and NA-P still appears high even if it was lower than for N-P. Do the authors mean to say … the relatively lower hydrophilicity of the A-P and NA-P?

Response to Comment 6

Sorry for the confusion and thank you for your comments. We meant by relatively lower hydrophilicity (A-P and NA-P relatively lower than N-P). Hence, we agree with you and now we have changed sentence accordingly.

Comment 7

Despite the superior hydrophilicity of the N-P samples compared with the A-P and NA-P samples, significant differences (p > 0.05) were not observed between the test groups in terms of the number of attached cells, which may have been related to limitations of the in vitro tests.

Which quantitative data does this refer to? Fig. 4 only shows qualitative data (images) of cell attachment. Please clarify how these images were quantitatively evaluated and show the relevant data.

Response to Comment 7

Again, sorry for the confusion and lack of information. We now have included new Figure 5, which is the results of quantitative analyses from 4 hours attachment following test and control exposure of NTAPPJ.

Minor Comments

The sequence of primers used in the qRT-PCR should be provided or referenced accordingly.

Minor spelling mistakes

- 2.4. Cell Proliferation, Alkaline Phosphatase Acitivy, and Osteogenic Differentiation on on Titanium Before and After NTAPPJ Exposure

= 2.4. Cell Proliferation, Alkaline Phosphatase Activity, and Osteogenic Differentiation on Titanium Before and After NTAPPJ Exposure

The C-P samples that were not subjected to the NTAPPJ treatment possess carbohydrates exhibited a lower degree of..

= The C-P samples that were not subjected to the NTAPPJ treatment and possess carbohydrates exhibited a lower degree of…

Therefore, the absorption of fibronectin is an important initial step in the osteogenic differentiation of hMSCs, and it was investigated by via qRT-PCR analyses of FAK gene expression in the hMSCs cultured on the samples for 6 hours in basic medium

= either: … investigated by qRT-PCR… or …investigated via qRT-PCR…

Response to Minor Comments

Thank you for your valuable comments and corrections. Primer sequence has been referenced accordingly. Also, all of corrections are made in accordance to the comments. Thank you once again for your review and kind help.

Reviewer 2 Report

Dear authors

Many thanks for your paper. It brings a lot of detailed studies in the field of plasma treatment of titanium surface and its applicability for selected cells grove. This is really interesting. Unfortunately, paper needs careful rearrangement and addition of missing figures that it was impossible to comment. So, below is list of the comments. I hope that they will help you for the paper improvement to acceptable form.

List of comments:

Abstract –­­­ Why -P is added into acronyms of samples? There is no reason for this. Also NA for humid ammonia is non-logic, HA will be more relevant.

Is it any reason for use of capital letters in sections’ titles?

Fig.1 – photos are not necessary. It will be better to enlarge schemes and their fonts. Use proper symbol for resistor (rectangle) in scheme A and add nitrogen flow controller/meter in scheme C.

Add detail information about experiment, this section is totally missing at proper place. It is given behind results and discussion but scheme is at proper place. Thus it is not readable.

SEM images can be omitted because they bring no new information and also there is not clear their scale. The information about contact angle is sufficiently given in text, only the experimental uncertainties must be added.

The contact angle changes during the aging must be commented in text and also it will be very useful if these results will be supported by explanation why is measured what is measured.

Figure 3 – the x-axis should be shorter to better show peaks. Mark groups directly to peaks instead the legend for better understanding. Fig. 3d should be transferred into table and extended by uncertainty information for each value.

Line 139 – what is difference between C-P and test group? Is the same or they are different? There is no information above about test group.

Fig. 4 – add scale bar size directly into images for better readability because it i different in each raw of figure.

2.4 – in section title double on on is used

Figures 5 and 6 are missing

Line 190-194 – there is nothing visible with respect to the Ti surface morfology from SEM images. May be, something will be visible using AFM.

Fig. 7 – left small schemes can be omitted. The detailed description should be given in text instead the figure caption.

Line 329 – specify ultrasonic bath temperature.

4.2 – calculation of applied energy simply by multiplying of voltage and current is non-correct. What is voltage and current waveform? Is system pulsed or not? System seems be similar to the dielectric barrier discharge, thus energy should be calculated by proper way using procedures given by Wagner et all in Vacuum.

4.3 – droplets seem be rather big, more common is use of 2 microliter drops that are measured immediately to avoid water evaporation. Half minute seems be too long delay. How many repetitions were used for these measurements? Samples, if they are treated at center without any scanning, are probably non-homogeneous, so the contact angles will be very sensitive on droplet position with respect to the center of the treated area.

Line 410 – number is not 1x104 but 1x10^4 of 1e4

Author Response

Reviewer #2

General Comments

Many thanks for your paper. It brings a lot of detailed studies in the field of plasma treatment of titanium surface and its applicability for selected cells grove. This is really interesting. Unfortunately, paper needs careful rearrangement and addition of missing figures that it was impossible to comment. So, below is list of the comments. I hope that they will help you for the paper improvement to acceptable form.

Response to General Comment

Thank you for your helpful comments. We now have carried out careful rearrangements and added missing figures. We hope that such improvement have resulted in the final manuscript in acceptable form.

Comment 1

Abstract –­­­ Why -P is added into acronyms of samples? There is no reason for this. Also NA for humid ammonia is non-logic, HA will be more relevant.

Response to Comment 1

We agree that -P may not be necessary and sorry for the confusion. However, if we use acronyms with only single letter such as A or P, this may cause confusion with some of figure letters we use. Hence -P has been added. Also, NA stands for mixture of nitrogen and ammonia as nitrogen gas has been used for humidified ammonia. This has been stated in Materials and Methods as;

Three different gases were used: nitrogen, air (compressed air), and humidified ammonia. In the case of humidified ammonia, special settings were considered involving the passage of nitrogen through the 0.1 M ammonia solution (Duksan Pure Chemical, Ansan, Korea) at a rate of 1 L/min (Fig. 1c).

Comment 2

Is it any reason for use of capital letters in sections’ titles?

Fig.1 – photos are not necessary. It will be better to enlarge schemes and their fonts. Use proper symbol for resistor (rectangle) in scheme A and add nitrogen flow controller/meter in scheme C.

Response to Comment 2

Thank you for your input and questions on the manuscript. We have used capital letter in sections’ title in accordance to other papers published in International Journal of Molecular Sciences. We will change the capital letter if the format is not correct.

Also, we now have modified Figure 1 in accordance to the comment.

Comment 3

Add detail information about experiment, this section is totally missing at proper place. It is given behind results and discussion but scheme is at proper place. Thus it is not readable.

SEM images can be omitted because they bring no new information and also there is not clear their scale. The information about contact angle is sufficiently given in text, only the experimental uncertainties must be added.

The contact angle changes during the aging must be commented in text and also it will be very useful if these results will be supported by explanation why is measured what is measured.

Response to Comment 3

Sorry about confusion with information about experiments. We have added Materials and Method after Discussion due to guidance on manuscript by this journal.

Figure 2 is now deleted in accordance to the comment and only changes in contact angle has been used for Fig. 2.

In addition, we now have commented on the new Fig. 2 regarding aging graph.

Comment 4

Figure 3 – the x-axis should be shorter to better show peaks. Mark groups directly to peaks instead the legend for better understanding. Fig. 3d should be transferred into table and extended by uncertainty information for each value.

Line 139 – what is difference between C-P and test group? Is the same or they are different? There is no information above about test group.

Fig. 4 – add scale bar size directly into images for better readability because it i different in each raw of figure.

Response to Comment 4

Thank you for your comment. We agree with your comment and Figure 3 has now been modified in accordance to the comment. The Figure 3d has been kept as the XPS results are based on single measurement and therefore there is no uncertainty information. This has now been clarified in Results.

In terms of test group, we considered that N-P, A-P and NA-P would be a test group. Sorry for this confusion and this is now clarified in Table 1.

We have now added scale bar size number directly onto Figure 4.

Comment 5

2.4 – in section title double on on is used

Figures 5 and 6 are missing

Line 190-194 – there is nothing visible with respect to the Ti surface morfology from SEM images. May be, something will be visible using AFM.

Fig. 7 – left small schemes can be omitted. The detailed description should be given in text instead the figure caption.

Line 329 – specify ultrasonic bath temperature.

Response to Comment 5

Thank you for your careful reading of this manuscript and sorry for such mistake.

We now have deleted ‘on’ that was double on 2.4

Figure 5 and 6 are now added, and sorry for forgetting these.

We now have deleted SEM images in accordance to your previous comment.

Left small schemes are now omitted from Figure 7 (now Figure 8).

Ultrasonic bath temperature has been stated.

Comment 6

4.2 – calculation of applied energy simply by multiplying of voltage and current is non-correct. What is voltage and current waveform? Is system pulsed or not? System seems be similar to the dielectric barrier discharge, thus energy should be calculated by proper way using procedures given by Wagner et all in Vacuum.

4.3 – droplets seem be rather big, more common is use of 2 microliter drops that are measured immediately to avoid water evaporation. Half minute seems be too long delay. How many repetitions were used for these measurements? Samples, if they are treated at center without any scanning, are probably non-homogeneous, so the contact angles will be very sensitive on droplet position with respect to the center of the treated area.

Line 410 – number is not 1x104 but 1x10^4 of 1e4.

Response to Comment 6

Thank you for your comments and sorry for the misunderstanding. We agree with your comment regarding power calculation and the values are now deleted to avoid confusion, and as the values would not be crucial in this manuscript.

The measurement of contact angle was based on previous study with regard to the volumes of liquid and also time of measurements. Hence, the reference is now stated to clarify the possible confusions.

Finally, the number has been modified, and we are sorry for such mistake.

Once again, thank you for your kind help and input to this manuscript.

Round 2

Reviewer 2 Report

Dear authors. Many thanks fot the improvements. The current version of your paper seems be OK.

Author Response

Dear authors. Many thanks fot the improvements. The current version of your paper seems be OK.

-> Thank you very much for your review and encouraging words.